# OpenReview forum: "Measuring Free-Form Decision-Making Inconsistency of Language Models in Military Crisis Simulations"
_ICLR.cc/2025/Conference — Submitted to ICLR 2025_

### Official Review · Reviewer_xYfm · 2024-10-31

**Soundness:** 2
**Presentation:** 3
**Contribution:** 2
**Rating:** 5
**Confidence:** 4

**Summary:**

This paper proposes to examine how reliable LLMs are in high stakes decision making situation. For this, the authors conduct crisis simulations with free form responses and using bertscore and measure the inconsistencies across 5 LLMs. Across experiments conducted, this study shows that there inconsistencies even after fixing parameters like conflict anonymization and temperature, and show that prompt variations can lead to greater inconsistency than temperature adjustments.

**Strengths:**

- the paper is well written and the methodology is easy to follow
- with the increasing use of LLMs in different areas, the focus on high-stakes decision-making contexts is timely and of importance

**Weaknesses:**

- The framework for measuring inconsistency uses only BertScore. This potentially limits the evaluation setting to the discrepancies found through semantic similarity missing other forms of inconsistency.
- There is no human evaluation or correlation to human judgement of the inconsistency score.
- I understand that the objective of this study is to probe LLMs for inconsistency in this particular context. While this study underlines a problem, it does not suggest possible mitigations for the issue of inconsistency.

**Questions:**

Besides inconsistency, is the proposed evaluation framework able to highlights other types of errors or discrepancies?

---

> ### Author Response · Authors · 2024-11-17
> **Response to Reviewer xYfm**
>
> Thank you for your thorough evaluation of our work! We have addressed your main concerns below:
>
> ### *W1: Limitation to only evaluating with BERTScore*
> We agree that our comparisons with other metrics were not framed clearly. In Section 3, we discuss why we do not choose to approach the problem with n-gram inconsistency measures - these are typically not able to capture semantic similarities. At the end of Section 4.1, we (albeit unclearly), mention that we tested a metric based on bi-directional entailment clustering. We found that it was not able to sufficiently parse out any similarities between responses that were otherwise conceptually dissimilar. We maintain that our verification of the validity of our inconsistency metric in Section 4 provides evidence that any sufficiently robust metric will yield very similar results (up to scaling/inversion). Part of the contribution is to verify that BERTScore is in fact trustworthy for measuring inconsistency in a generalized question-answering setting.
>
> ### *W2: No human evaluation or correlation to human judgment*
> We kindly refer you to the first part of our general response titled "Further analysis and human evaluation for inconsistency score."
>
> ### *W3: No discussion of possible mitigations*
> We addressed this in the second part of our general response titled "Why we do not mitigate inconsistency."
>
> ### *Q1: Can the evaluation framework highlight other types of discrepancies?*
> Thank you for this question! We showed that our metric can reliably distinguish between "safe" and "unsafe" responses to mental health emergencies in an additional experiment (we expand on details in the general response). Thus, we hope future work can build on this evaluation framework and study ways to automatically evaluate free-form responses of agentic systems in various settings. For example, a researcher may define categories under which the responses fall under and subsequently use our evaluation framework and well-defined comparisons to categorize responses to expose conceptual discrepancies. Broadly, we hope future work can use this framework to explore inconsistency-mitigating decoding strategies surrounding LM self-consistency, automatically categorize free-form text into conceptual categories, or further study LM behavior in further high-stakes domains (e.g. law, government).
>
> We hope our new experiments and arguments have addressed your concerns! We look forward to hearing your thoughts and are happy to conduct new experiments to address any further concerns. Again, thank you for your time and constructive feedback!

---

> > ### Comment · Reviewer_xYfm · 2024-11-22
> >
> > I thank the authors for the thourough work in addressing the concerns raised. I still think that more in depth analysis and experimental comparison is need. In the current state, it is difficult to position the nolvelty and siginificance. Additionally, I feel like the default should temperature should have been T=0 and then additional results given for T=1 motivated by the reasoning given in the rebuttal.

---

> > > ### Author Response · Authors · 2024-11-23
> > > **Further Response to Reviewer xYfm**
> > >
> > > Thank you for your response and further insight! We address your further concerns below:
> > >
> > > ***more in depth analysis and experimental comparison is needed***
> > >
> > > As mentioned, we provide empirical evidence that a method based on bi-directional entailment clustering does not suffice. Our evaluation framework is metric agnostic - as long as another metric is able to capture inconsistency similar to how BERTScore does, one may do the same evaluation. We absolutely agree with the idea that other metrics of this capacity likely exist. Do you think it would strengthen the paper to mention this? We maintain that this work is important in terms of advancing beyond maximizing human correlation coefficients - BERTScore may be used in inconsistency evaluation that would otherwise be meaningless without our tests. Our additional experiments on wargame specific synthetic responses strengthen the task-specific case for the robustness of the metric. Additionally, our experiments on mental health related responses strengthen the case for the generalizability of the metric to different high-stakes application domains. We would greatly appreciate it if you could clarify the type of experimental comparison that you would like to see. Thank you!
> > >
> > > ***it is difficult to position the novelty and significance***
> > >
> > > We reiterate that our contribution is rooted in the demonstration of the existence of a metric that can robustly capture inconsistency in an explicit QA setting, ignoring linguistic variations that do not affect semantic meaning. That is, we establish an otherwise unestablished necessary precondition for evaluation of this type. Evaluation of LMs in high-stakes contexts is of great significance currently given the rapid adoption of LMs into various domains. We establish their proliferation into military spheres, a domain that can give rise to particularly catastrophic consequences under misguided decision-making [1, 2, 3]. We would view it as a great loss to the community to reject work that addresses high-impact social issues on the basis of a lack of novelty given that we have shown that an existing metric works for the type of evaluation we are conducting. We still maintain the novelty in our work in validating BERTScore and applying our inconsistency evaluation to a new domain - free-form military crisis decision-making.
> > >
> > > ***Additionally, I feel like the default should temperature should have been T=0 and then additional results given for T=1 motivated by the reasoning given in the rebuttal.***
> > >
> > > This is an interesting idea that we would like to explore. Do you think it would be more meaningful to, say, put Section 6 (prompt variation experiments at T = 0) above Section 5 (evaluation at T=0)?
> > >
> > > Please let us know if there are any further actionable steps we can take to warrant a score increase. Again, thank you for your thoughtful evaluation of our work!
> > >
> > > [1] National Security Archive. False Warnings of Soviet Missile Attacks Put U.S. Forces on Alert in 1979-1980, 2020. URL https://nsarchive.gwu.edu/briefing-book/nuclearvault/2020-03-16/false-warnings-soviet-missile-attacks-during1979-80-led-alert-actions-us-strategic-forces. \
> > > [2] Geoffrey Forden, Pavel Podvig, and Theodore A Postol. False alarm, nuclear danger. IEEE Spectrum, 37(3):31–39, 2000. \
> > > [3] EUCOM History Office. This Week in EUCOM History: January 23-29, 1995, 2012. URL https://web.archive.org/web/20160105033448/http://www.eucom.mil/ media-library/article/23042/this-week-in-eucom-history-january23-29-1995.

---

> > > > ### Comment · Reviewer_xYfm · 2024-11-25
> > > >
> > > > Thanks for clarifying further.
> > > > To address the first point, the paper lacks a clear motivation for the testing of bi-directional entailment clustering, finding it doesn's work and using BertScore. Explaining this point would help in my opinion.
> > > >
> > > > Then, please, correct me if I'm wrong about this point. While I understand that the study posits that inconsistencies arise from a lack of semantic coherence, it's unclear why semantic consistency should be expected at a temperature of T=1 for this task. Is that an LLM issue? A task issue?
> > > >
> > > > I agree with your rebuttal to my second point.
> > > >
> > > > > This is an interesting idea that we would like to explore. Do you think it would be more meaningful to, say, put Section 6 (prompt variation experiments at T = 0) above Section 5 (evaluation at T=0)?
> > > >
> > > > yes, I think this would help a lot.

---

> > > > > ### Author Response · Authors · 2024-11-25
> > > > > **Further Response to Reviewer xYfm**
> > > > >
> > > > > Thanks for the response and the questions!
> > > > >
> > > > > ***Lack of clear motivation for testing of bi-directional entailment clustering***
> > > > >
> > > > > Thank you for pointing this out - we agree that we framed this unclearly. To clarify, we mentioned this to demonstrate that we considered potential metrics beyond just blindly choosing and verifying BERTScore for our analysis. We found that it was not able to sufficiently extract any similarities between texts that are otherwise conceptually dissimilar. Thus, we obtained inconsistency scores that were extremely high (~0.9), making comparison and evaluation meaningless. When testing BERTScore, we found that, unlike bi-directional entailment clustering, it was able to robustly capture inconsistency (and consistency), so we focused our analysis using this metric. We will clarify this point in the revised pdf.
> > > > >
> > > > > ***Unclear why semantic consistency should be expected at T=1***
> > > > >
> > > > > Thank you for this question! I think a good way of thinking about this would be to consider how an LM would behave on a task that it does well at. We would expect, despite sampling at T=1, for the LM to be able to give consistent answers and reasoning because it is more “confident” in its answers. This interpretation has been successfully implemented in the literature for hallucination detection and mitigation [e.g., 1, 2]. We do not make any claims about LM hallucination because there is no notion of ground-truth in military decision-making. However, the idea that putting trust in an inconsistent, unconfident agent leading to volatile and unpredictable decision-making still holds. Of course as you rightly point out, we should not expect perfect consistency at T= 1. But, if LMs were “good” and “confident” at military crisis decision-making, we should expect to see much more semantic consistency than we observe, even at T = 1. In fact, we still observe high levels of inconsistency even at a lower T = 0.2 (see figure 6 in the pdf).
> > > > >
> > > > > We believe that inconsistency is likely both an LM issue and task issue. It is an LM issue in the sense we described above. It is a task issue in that the task itself calls for open-ended decision-making, likely opening the door for more unpredictable responses. But we reiterate that our focus on free-form responses is well motivated by real world tests conducted by militaries worldwide and products being developed by private companies, as mentioned in the paper.
> > > > >
> > > > > ***Switching section order***
> > > > >
> > > > > Thank you for the feedback on this! We will update the section order in the revised pdf.
> > > > >
> > > > > [1] Potsawee Manakul, Adian Liusie, and Mark Gales. SelfCheckGPT: Zero-resource black-box hallucination detection for generative large language models. In Proceedings of the 2023 Conference on Empirical Methods in Natural Language Processing, pp. 9004–9017. Association for Computational Linguistics, 2023b.\
> > > > > [2] Sebastian Farquhar, Jannik Kossen, Lorenz Kuhn, and Yarin Gal. Detecting hallucinations in large language models using semantic entropy. Nature, 630(8017):625–630, 2024.

---

### Official Review · Reviewer_cN3C · 2024-11-02

**Soundness:** 3
**Presentation:** 4
**Contribution:** 2
**Rating:** 3
**Confidence:** 5

**Summary:**

This paper presents an investigation of the consistency of LLM responses in a military strategy scenario. Authors invert the BERTScore measure of semantic similarity to compute an inconsistency score, which they validate in a synthetic free-form QA setting based on Truthful QA. In the experiments, the paper shows that LLMs' answers are generally quite inconsistent in both types of generations for their scenario (initial, continuations). They also explore the effect of temperature on inconsistency, as well as the effect of prompt variations.

**Strengths:**

- I appreciate the paper's main goal of investigating LLM inconsistencies in high-stake military scenarios
- I liked that the authors performed a validation of the inconsistency score using synthetic data with TruthfulQA, which also allows readers to calibrate on what score values mean.
- The main experiments were well executed.
- I liked the investigations into the prompt variations.
- I also appreciated the disclaimer at the end of the introduction.

**Weaknesses:**

- The paper's main weakness is the conceptual problem definition.
    - I feel like in realistic high-stakes settings, the temperature should probably be set to 0, which is similar to greedy decoding. Authors should probably focus their experiments on that temperature, though I'm expecting very low inconsistency. More importantly, authors should make a clearer argument for why t=0 should not be used in these high-stakes settings, or why t>0 should be studied.
     - Concretely, I feel like section 6 was most relevant to the realistic way that LMs should be used. I wish authors had expanded on such experiments, possibly exploring various types of rephrasing and exploring exactly how the inconsistencies were affected.
- Second, I feel like the paper's results might be limited in terms of generalizability due to there being only one wargame scenario considered. I understand that this is a relevant vignette, but it would be very interesting to have at least 3-5 high-stakes scenarios (either all wargame or some other high-stakes domains like healthcare). This would ensure that the results aren't an artefact of the topics or domain chosen in the one scenario.
- Third, I felt like the paper could use more in-depth analyses w.r.t. how model responses are inconsistent. The measure at hand is quite coarse-grained, and might not be able to capture more nuanced consistent/inconsistent outputs (e.g., LLM outputs offering two alternatives, one of which is similar between two outputs). Given the specific and high-stakes nature of the scenario, it'd be really useful to have more insights into how the outputs differ, as currently, the coarse-grained measure yields very little information about how inconsistencies should be mitigated (other than lowering the temperature).
- Missing citations: http://arxiv.org/abs/2203.12258, https://arxiv.org/abs/2311.00059, https://arxiv.org/abs/2310.11324
- L204: Authors mention a bi-directional entailment clustering method, but without more details, it seems very confusing why the authors mentioned that... I would remove that sentence or specify why they needed to test that method and why they didn't include the results in the final main text.

**Questions:**

NA

---

> ### Author Response · Authors · 2024-11-17
> **Response to Reviewer cN3C**
>
> Thank you for your thoughtful review and constructive feedback! We address your concerns below:
>
> ### *W1: Why not temperature T = 0.0?*
> We kindly refer you to the first part of our general response under the section titled “Why we evaluate LMs at temperature T > 0.” We will do our best to meet your wishes exploring the expansion of section 6.
>
> ### *W2: Limited Generalizability*
> We kindly refer you to the first part of our general response under the section titled “Regarding Generalizability Concerns.”
>
> ### *W3: More in-depth analysis on inconsistency, coarse-grained measure*
> We address this concern in the first part of our general response under the section titled "Further analysis and human evaluation for inconsistency score." Additionally, we address why we do not pursue mitigating inconsistency directly in this work in the second part of our general response under the section titled "Why we do not mitigate inconsistency."
>
> ### *W4: Missing Citations*
> Thank you for pointing out the missing citations! We will add these to the revised submission. The first and third one seem particularly pertinent to our prompt sensitivity experiments!
>
> ### *W5: Mention of bi-directional entailment clustering*
> Thank you for pointing this out - we agree this is unclear. We mentioned this to demonstrate that we considered other potential metrics beyond BERTScore for our analysis. We did not include these results in the main text because we found that this metric was unable to sufficiently extract any similarities between texts that are otherwise conceptually dissimilar. Thus, we obtained inconsistency scores that were extremely high across all settings and models (~0.9), making comparison and evaluation meaningless. We found that BERTScore more robustly captured inconsistency, and so we focused our analysis using this metric. We will clarify these points in the revised pdf.
>
> We hope our additional experiments and clarifications have addressed your concerns, we would be grateful if the reviewer re-evaluate our work. Again, thank you for your time and constructive feedback! Please let us know if you would like us to run additional experiments or if you have any additional concerns.

---

> > ### Comment · Reviewer_cN3C · 2024-11-22
> >
> > Thank you for the rebuttal! Most weaknesses are addressed in my opinion, except for the main one which prevents me from altering my score.
> >
> > # W1
> > I find it hard to believe that military people would no be able to have deterministic AI systems, this feels very strange to me. I cannot simply trust that you did surveys with many military people, and will require citations or evidence that that is the case. If the authors aim to make a claim that military people require a diverse set of outputs from LLMs (with citations), then that would warrant taking a temperature T>0. However, the current paper focuses on inconsistency of LLM outputs, not the tradeoff between consistency and diversity.

---

> ### Author Response · Authors · 2024-11-23
>
> We appreciate your feedback and understand your skepticism regarding the use of non-deterministic AI systems (T > 0) in military settings. We want to clarify our position and provide extensive evidence to support our claims, although they may seem unintuitive for standard LM evaluations. Predictability in military decision-making is universally considered to be a significant vulnerability. Adversaries capable of anticipating actions may exploit consistent patterns to undermine strategies. Military doctrines and strategic studies emphasize the importance of unpredictability to maintain a tactical advantage:
>
> - Game Theory and Mixed Strategies: In competitive and adversarial scenarios, game theory advocates for mixed strategies, which involve randomizing choices to prevent opponents from predicting actions [1][2]. This concept is crucial in military applications to avoid being outmaneuvered by adversaries who might exploit predictable decision patterns.
>
> - Military Doctrine Emphasizing Flexibility and Adaptability: Renowned historical military strategists like Sun Tzu and Clausewitz have underscored the importance of adaptability and unpredictability in warfare to outsmart opponents [3][5]. Modern military doctrines continue this emphasis: The U.S. Army's Operational Art Primer highlights the need for commanders to employ creativity and adaptability, integrating ends, ways, and means across the levels of war [4]. Deception and unpredictability are considered essential for achieving strategic surprise and maintaining operational security [6].
>
> Given these principles, deploying deterministic AI systems with T = 0 could introduce risks due to their predictable outputs in case of cybersecurity failures. In cybersecurity threats or espionage scenarios, adversaries could exploit this predictability to anticipate and counteract military strategies. While our paper focuses on the inconsistency of LLM outputs, understanding the trade-off between consistency and diversity is crucial in high-stakes settings. Setting the temperature T > 0 introduces controlled randomness, aligning with the strategic need for unpredictability.
>
> Nevertheless, we also back up our findings with an extensive range of variations at T = 0. As discussed in our general rebuttal, both types of experiments (T>0 and T =0) are crucial to meaningfully comparing free-form response inconsistency of LM decision-making. Would you prefer if we re-order the experiments in the paper to start with section 6 and then do section 5?
>
> Our research highlights the importance of addressing this inconsistency to ensure reliable yet unpredictable decision-making support. By examining how LLMs behave under different temperature settings, we provide crucial insights into limitations to safe deployments.
>
> We will revise our paper to include these citations and clarify our argument regarding the strategic necessity of employing LLMs with T > 0 in such high-stakes settings. Thank you for your valuable feedback. We believe this clarification strengthens our paper and addresses your concerns.
>
> References:
> [1] Osborne, M. J., & Rubinstein, A. (1994). A Course in Game Theory. MIT Press.
>
> [2] Myerson, R. B. (1991). Game Theory: Analysis of Conflict. Harvard University Press.
>
> [3] Sun Tzu. (5th Century BCE). The Art of War.
>
> [4] Sweeny, P. (2010). Operational Art Primer. U.S. Department of the Army.
>
> [5] Howard, M., & Paret, P. (1976). Clausewitz: On War. Princeton University Press.
>
> [6] Barlow, R. E. (2006). "Deception and Unpredictability in Military Operations". Naval War College Review, 59(1), 43–53.

---

> > ### Comment · Reviewer_cN3C · 2024-11-25
> >
> > Hi there.
> >
> > Thanks for the clarifications. Yet, I still find that your claims are contradictory.
> >
> > In the paper, you mention that LLMs in high-stakes settings "require consistent, reliable decision-making" (058). You then also say "delegating trust to an inconsistent agent can lead to unpredictable decision-making, which is a cause for concern" (065). The discussion of results further frames inconsistency as bad, e.g., "Encouragingly, we find that lexical substitution and syntactic restructuring generate the least inconsistency" (209).
> >
> > However, in the rebuttal, you are now saying that predictability is an issue in military settings and call for LLMs that should "ensure reliable yet unpredictable decision-making support" but that is not what the investigations in your paper actually do. They do not disentangle reliability from unpredictability (yet that would be a really interesting but challenging thing to try to measure).
> >
> > I think the whole paper would require substantial reframing in order to resolve this contradiction, and possibly require some additional experiments to support the new framing of "consistency = predictability, which is bad"

---

> ### Author Response · Authors · 2024-11-26
> **Further Response to Reviewer cN3C**
>
> Hello! Thank you again for your thoughtful engagement. We understand how you view our claims as contradictory. However, I am afraid there may have been a misunderstanding.
>
> For maximum clarity in this response, we will say *external unpredictability* when referring to being unpredictable in the eyes of the adversary. We will say *internal unpredictability* when referring to being unpredictable in the eyes of the institution deploying the LM for use in their own operations. Our arguments in the rebuttal focused on external unpredictability. What we established in our previous rebuttal is that militaries are incentivized to avoid setting T = 0 to avoid being externally predictable as a result of deterministic responses.
>
> What we focus on in our paper is the notion of internal unpredictability (and likely why you see a contradiction in our framing). For example, in (058), we are saying LMs in high-stakes settings should be required to be internally reliable. In (068), we are saying that delegating trust to an inconsistent agent can lead to being internally unpredictable. This is a cause for concern for many reasons [e.g., 1, 2, 3]. So, when we say “ensure reliable yet unpredictable decision-making support” we refer to internal reliability while being externally unpredictable. What we ultimately show in our paper is that LMs are highly inconsistent at T > 0, which, while making them externally unpredictable, also makes them internally unpredictable and unreliable.
>
> So, what we ultimately argue is that if militaries are to deploy LMs, they must find a balance between the desire for external unpredictability and internal predictability/reliability. What we show in the paper is that, in the context of LM automated or augmented decision-making, these notions are inherently at odds. This tension is likely unsolvable (or at least extremely difficult to solve), calling into question the very deployment of LMs into military operations. To reiterate, nowhere in our paper do we make claims that LMs should be deployed into militaries while being simultaneously unpredictable and predictable. What we do show is that LMs, under the premises in which they are likely to be deployed, exhibit behavior that calls into question their internal reliability. This behavior is a cause for concern and can even lead to catastrophic consequences [4, 5, 6]. In fact, the final sentence of our paper (539) says that robust safeguards should be put in place to prevent unintended outcomes that may arise due to the deployment of LMs into military operations.
>
> Therefore, we do not believe there to be a contradiction in our framework. We hope this clears up the premise and motivation of the paper. It would be regrettable for this work to be rejected under a conceptual misunderstanding.
>
> In acknowledgement of your valid questions regarding our conceptual problem definition, we will add an entire section in the appendix of the revised pdf clarifying this our arguments in this discussion to motivate our experimental setup. Additionally, as Reviewer xYfm suggested, we will move section 6 (prompt sensitivity experiments at T = 0) above section 5 (evaluation at T = 1) to better frame the point of the paper.
>
> We are grateful to have engaged in this discussion with you, and hope that our arguments warrant a score increase in your eyes.
>
> [1] William N Caballero and Phillip R Jenkins. On Large Language Models in National Security Applications. arXiv preprint arXiv:2407.03453, 2024.\
> [2] Juan-Pablo Rivera, Gabriel Mukobi, Anka Reuel, Max Lamparth, Chandler Smith, and Jacquelyn Schneider. Escalation risks from language models in military and diplomatic decision-making. In The 2024 ACM Conference on Fairness, Accountability, and Transparency, pp. 836–898, 2024.\
> [3] Lamparth, M., Corso, A., Ganz, J., Mastro, O. S., Schneider, J., & Trinkunas, H. (2024). Human vs. Machine: Behavioral Differences between Expert Humans and Language Models in Wargame Simulations. Proceedings of the AAAI/ACM Conference on AI, Ethics, and Society, 7(1), 807-817.\
> [4] National Security Archive. False Warnings of Soviet Missile Attacks Put U.S. Forces on Alert in 1979-1980, 2020. URL https://nsarchive.gwu.edu/briefing-book/nuclearvault/2020-03-16/false-warnings-soviet-missile-attacks-during1979-80-led-alert-actions-us-strategic-forces.\
> [5] Geoffrey Forden, Pavel Podvig, and Theodore A Postol. False alarm, nuclear danger. IEEE Spectrum, 37(3):31–39, 2000.\
> [6] EUCOM History Office. This Week in EUCOM History: January 23-29, 1995, 2012. URL https://web.archive.org/web/20160105033448/http://www.eucom.mil/ media-library/article/23042/this-week-in-eucom-history-january23-29-1995.

---

### Official Review · Reviewer_McSh · 2024-11-08

**Soundness:** 3
**Presentation:** 4
**Contribution:** 2
**Rating:** 5
**Confidence:** 4

**Summary:**

This paper investigates the inconsistency of large language models (LLMs) in terms of ablations like sentence order and semantics when applied in war games. The authors focus on measuring free-form response variability using BERTScore to quantify inconsistencies. The findings indicate that the LLMs exhibit considerable inconsistency in their response, which raises concerns about their reliability in high-stake decision-making contexts.

**Strengths:**

1. The overall presentation is very clear and intuitive
2: The experimental setups are rigorous - experiments are run with multiple LLMs, variations in prompt scenarios, and statistical controls to examine the consistency across different levels of temperature settings and prompt structures.
3. Besides quantitative measures, the paper provides qualitative examples that provides valuable insights
4. The prompts are fully provided for ease of reproduction

**Weaknesses:**

1. The paper focuses on a very specific hypothetical military scenario. It’s also uncertain whether the observed inconsistency is unique to the wargame setup or would generalize to other critical decision-making applications. This might limit the generalizability to other high-stakes applications.
2. The paper’s main innovation centers on using BERTScore as an inconsistency measure, which may not offer significant novelty in approach.
3. The study also did not sufficiently compare this approach with other potential inconsistency measurements.
4. The choice of a default temperature setting of 1.0 in Section 5 may not be appropriate, as it introduces significant response inconsistency by design.
5. The comparison are limited to a few closed-source LLMs
6. While the study demonstrates that LLMs can produce inconsistent responses, it would be more impactful if it included strategies for reducing such variability.

**Questions:**

1. Why was the wargame scenario chosen as the primary setting for examining inconsistency in high-stake decision making? Do you believe the inconsistency findings would generalize to other types of critical scenarios, or are they specific to the military context?
2. In Section 5, why was a temperature of 1.0 chosen as the default setting?
3. Could this research potentially inform new methods to improve LLM consistency in decision-making contexts?

---

> ### Author Response · Authors · 2024-11-17
> **Response to Reviewer McSh**
>
> Thank you for the thoughtful feedback! We address your concerns below:
>
> ### *W1/Q1: Limited generalizability due to constrained evaluation setting*
> We kindly refer you to part 1 of our general response under the section titled “Regarding generalizability concerns.”
>
> ### *W2: Limited Novelty*
> We actually considered developing a new metric. However, as we show, BERTScore performs well in detecting inconsistency on decision-making tasks. We believe it is more impactful to the research community to show that an existing metric can generalize into a new task domain rather than develop a novel metric just for the sake of algorithmic novelty. We maintain that novelty comes from the verification that BERTScore is a reliable metric in decision-making/QA settings. Additionally, we are the first to study LM free-form inconsistency in the military domain. Reviewer xYfm recognizes the importance of this work given the societal implications surrounding the adoption of LMs into high-stakes domains.
>
> ### *W3: Lack of comparison with other inconsistency measurements*
> We agree that our comparisons with other metrics were not framed clearly. In Section 3, we discuss why we do not choose to approach the problem with n-gram inconsistency measures - these are typically not able to capture semantic similarities. At the end of Section 4.1, we (albeit unclearly), mention that we tested a metric based on bi-directional entailment clustering. We found that it was not able to sufficiently parse out any similarities between responses that were otherwise conceptually dissimilar. We maintain that our verification of the validity of our inconsistency metric in Section 4 provides evidence that any sufficiently robust metric will yield very similar results (up to scaling/inversion).
>
> ### *W4/Q2: Choice of temperature 1.0*
> We kindly refer you to part 1 of our general response under the section titled “Why we evaluate LMs at temperature T > 0.”
>
> ### *W5: Evaluation limited to few models*
> This is a good point. We'll run our experiments on more models, particularly on those that are open-source.
>
> ### *W6: We do not study inconsistency mitigation*
> We kindly refer you to part 2 of our general response under the section titled “Why we do not mitigate inconsistency.”
>
> ### *Q1: Why did we choose the wargame scenario?*
> We choose to focus our analysis on the wargame setting as this is a worrying real-world application of LMs where studying inconsistency will have grounded implications. We maintain the importance of our work in rigorously exposing any pitfalls of LMs given the sensitivity of the domain. As mentioned, we perform further experiments on chatbot responses pertaining to mental healthcare and find that inconsistency and our evaluation framework generalizes to another critical scenario.
>
> ### *Q3: Could this research potentially inform new methods to improve LLM consistency in decision-making contexts?*
> This is a fantastic question! As mentioned above, much of the contribution of this work is testing a metric that can robustly measure inconsistency, a necessary precondition to *mitigating* inconsistency. As we did when showing our metric can distinguish between "safe" and "unsafe" responses to mental health emergencies, we hope future work can build on this evaluation framework and study ways to automatically evaluate free-form responses of agentic systems in various settings. We hope future work can use this framework to explore inconsistency-mitigating decoding strategies surrounding LM self-consistency, automatically categorize free-form text into conceptual categories, or further study LM behavior in further high-stakes domains (e.g. law, government).
>
> We hope these new experiments and arguments have addressed your concerns! We look forward to hearing your thoughts and are happy to conduct new experiments to address any further concerns.

---

> > ### Author Response · Authors · 2024-11-26
> > **Official Comment from Authors**
> >
> > Dear Reviewer McSh, given that we are nearing the end of the author-reviewer discussion period, we wanted to send a friendly reminder that we'd love to hear your thoughts on our rebuttal! Please let us know if there are any remaining questions or concerns. If we addressed your previous concerns sufficiently, we'd be delighted if you'd consider raising your final score. Thank you so much again for your time and feedback!

---

### Author Response · Authors · 2024-11-17
**General Response (1/2)**

Dear reviewers and chairs,

Thank you to the reviewers for your constructive reviews and everyone for their consideration! Below, we address common concerns shared by all reviewers.

### *Why we evaluate LMs at temperature T > 0*
We have consulted academic scholars in international and national security throughout the course of this research. They said militaries cannot afford their systems to give deterministic decisions in case of cybersecurity failures leading to  predictable decision-making. Thus, we have strong reason to believe militaries will not set the temperature to 0.0. We agree this is unintuitive, and we will clarify this in the paper. We uphold that this fact makes the results all the more pressing - those implementing LMs in high-stakes settings should find a way to robustly parse through their inconsistency or scrap their use altogether.

Additionally, related works have analyzed LM inconsistency as a method for hallucination detection [e.g., 1, 2]. These works hinge on the assumption that high inconsistency levels imply low model confidence, and often set T = 1.0 for baseline experiments. Therefore, by setting T = 1.0, we are able to proxy a notion of model confidence and reliability when responding to fixed prompts. This establishes a necessary comparison point for the experiments in Section 6, where we set T = 0 and find that model inconsistency due to prompt variations when responding with T = 0 can lead to inconsistency levels comparable to model responses to fixed prompts at T > 0 (and up to 1.2 for some models).

Lastly, previous work has shown that there are limitations to greedy decoding [e.g., 3, 4, 5]. Thus, it is reasonable to expect that LMs be deployed at T > 0 even if it comes at the cost of less consistency.

### *Regarding generalizability concerns*
We see the reviewers’ concerns regarding generalizability and agree that applying our framework to more applications would strengthen our results. To this end, we run additional experiments on free-form responses of chatbots interacting with users in mental health emergencies using the public dataset from [6]. From that dataset, we use the responses from frontier closed-source models and open-source models (ChatGPT-3.5, ChatGPT-4, Mistral-7b, Llama2-7b, Llama2-13b, Claude-3-opus, Gemini) which are expert human labeled as either “safe,” “unsafe,” or “borderline”.

We find that models still exhibit high levels of inconsistency. Additionally, we find that our inconsistency metric is able to distinguish between safe and unsafe responses with statistical significance. We also find that borderline responses were significantly closer to safe responses than unsafe responses. We will add this evaluation to the appendix in the revised pdf. These results show that our inconsistency evaluation framework generalizes to a different context and different models not present in our initial analysis.

### *Further analysis and human evaluation for inconsistency score*
We have conducted another set of new experiments to test how our inconsistency score behaves under synthetic, on-distribution ablations specific to the experimental scenario. We selected example responses and changed between one to five “actions” conveyed in the response, while keeping all other text (including other actions) in the response the same. Keeping the other actions unablated and text exactly identical creates more stringent evaluation conditions for the inconsistency score as compared to the true conditions present in the main experiments.

We find 1) inconsistency increases approximately linearly as we change additional actions, 2) Changing just 2 actions while keeping the rest of the text exactly the same yields inconsistency scores indicative of substantial semantic difference. Because of the stringent test conditions, the observed inconsistency scores are a lower bound to those we would see when evaluating on the true dataset. We will add these results to the appendix in the revised pdf.

Additionally, we qualitatively evaluated many of the sample responses and their corresponding inconsistency scores. We will include a representative sample of response pairs and their corresponding inconsistency scores in the appendix of the revised pdf. Finally, as mentioned in the paper, we use BERTScore based on the DeBERTa xlarge model, which achieved a pearson correlation with human judgment of 0.7781 [7], the highest of all supported models. We will be sure to include this in the revised pdf as well.

*Continued*

---

> ### Author Response · Authors · 2024-11-17
> **General Response (2/2)**
>
> ### *Why we do not mitigate inconsistency*
> We actually considered mitigating inconsistency over the course of the research. However, prior to finding mitigations for inconsistency, it was necessary to either develop or verify a metric that robustly captures inconsistency in general decision-making contexts as, to our knowledge, one does not yet exist for such settings. Thus, we focus our work on finding and verifying such a metric and study how such measurements are affected by temperature and prompt sensitivity. We show that for any method that aims to improve inconsistency, it is necessary to evaluate the performance of the metric at different temperatures and across prompt variations.
>
> While having a metric is a necessary precondition, there are other challenges associated with mitigating free-form inconsistency, particularly in high-stakes applications. In this work, it would not be impactful to, say, fine-tune models to reduce inconsistency because this would require one set of arbitrary strategic preferences over others. These naturally meaningfully differ across individuals, societies, and cultures. Additionally, conducting such fine-tuning would raise severe ethical concerns surrounding the enabling of automated military decision-making. For high-stakes applications in general, our work demonstrates that meaningful evaluations and governance efforts of safety-critical deployment settings must go beyond just capability evaluations and consider inconsistency due to both sampling temperature and prompt sensitivity.
>
> Thank you to all reviewers and chairs for considering our submission for ICLR. The reviews were instrumental in improving our work.
>
> #### *References*
> [1] Potsawee Manakul, Adian Liusie, and Mark Gales. SelfCheckGPT: Zero-resource black-box hallucination detection for generative large language models. In Proceedings of the 2023 Conference on Empirical Methods in Natural Language Processing, pp. 9004–9017. Association for Computational Linguistics, 2023b.\
> [2] Sebastian Farquhar, Jannik Kossen, Lorenz Kuhn, and Yarin Gal. Detecting hallucinations in large language models using semantic entropy. Nature, 630(8017):625–630, 2024.\
> [3] Ari Holtzman, Jan Buys, Maxwell Forbes, Antoine Bosselut, David Golub, and Yejin Choi. Learning to write with cooperative discriminators. In Proceedings of the Association for Computational Linguistics, 2018.\
> [4] Xinyun Chen, Renat Aksitov, Uri Alon, Jie Ren, Kefan Xiao, Pengcheng Yin, Sushant Prakash, Charles Sutton, Xuezhi Wang, and Denny Zhou. 2023b. Universal self-consistency for large language model generation. ArXiv, abs/2311.17311.\
> [5] Prabhu, Sumanth. "PEDAL: Enhancing Greedy Decoding with Large Language Models using Diverse Exemplars." arXiv preprint arXiv:2408.08869 (2024).\
> [6] Declan Grabb, Max Lamparth, and Nina Vasan. Risks from Language Models for Automated Mental Healthcare: Ethics and Structure for Implementation. In First Conference on Language Modeling, 2024.\
> [7] BERTScore. BERTScore. https://github.com/Tiiiger/bert score, 2020. [Online; accessed 30-September-2024].

---

### Author Response · Authors · 2024-11-27
**Summary of Revisions**

We thank the reviewers for their continued engagement with our work! In response to the original comments and points brought up during discussion, we have updated the paper with a few revisions.

In the revised version, you will see text in *red* and text in *blue*. The text in *blue* is content that was exactly present in the original version (bar minor rephrasing) that was moved into a new section in the revision. The text in *red* is new content or content that we majorly changed.

**Summary of major changes:**
1. We added a discussion of our new experiments measuring inconsistency on chatbot responses to mental health crises. The full discussion is provided in Appendix E, with mentions in Section 4.2 and 7. This aims to address reviewer concerns regarding whether high inconsistency generalizes to other critical task domains, as well as whether our inconsistency score (and generally, BERTScore) can be used in other evaluation frameworks in other contexts.
2. We added a discussion of our more fine-grained analysis of BERTScore on synthetic, wargame-specific responses. The full discussion is provided in Appendix C, with a mention in Section 4.3 We also added a representative sample of response pairs taken from the experiments  alongside their inconsistency scores to provide further human evaluation (Appendix D). Finally, we added that the BERTScore model we implement achieved a Pearson correlation with human judgment of 0.7781 as found by the original authors of the BERTScore paper to Section 3. This aims to address reviewer concerns pertaining to the interpretation and validity of the inconsistency score.
3. We re-order Section 5 and Section 6 as recommended by Reviewer xYfm. That is, we moved our prompt sensitivity experiments above our evaluation at T = 1. This is in response to reviewer concerns regarding our conceptual problem definition and belief that the paper would be better framed if T = 0 was used as the default temperature. To that end, in the new Section 6, we write a bit about *why* we believe T = 1 to be well-motivated and worth evaluating. Because we acknowledge that is an unintuitive concept, particularly to those removed from national security circles, we provide a full discussion of this motivation in Appendix G.
4. We demote the mention of bi-directional entailment clustering to a footnote so as to not overstate its importance to the present work and maintain the focus on BERTScore and our contributions pertaining to the validation of it for inconsistency evaluation. We add a sentence explaining why we do not include it in the main body. We maintain the full discussion of our tests with the method and why exactly we do not include it in Appendix F. This is in response to reviewer recommendations to clarify the mention of bi-directional entailment clustering and to better position an aspect of our contribution.

We also made some minor changes. For example, to accommodate the re-ordering of sections, we slightly change the abstract and introduction to ensure that our presentation stays strong and true to the present paper. We also added some missing citations, including those recommended by Reviewer cN3C.

We appreciate all the reviewers' feedback and time! We believe our paper is much stronger as a result. We have conducted additional experiments and addressed all concerns raised in the reviews to the best of our ability. We would greatly appreciate it if the reviewers could consider revising their scores in light of these updates or kindly provide further details if any concerns remain unaddressed.

Sincerely,\
The Authors

---

### Meta-Review · Area_Chair_5b6x · 2024-12-21

**Metareview:**

**Summary:**

The authors analyze how previously observed inconsistency in outputs of LLMs influences their behavior during military decision-making and mental health simulations. They use BERTScore for identifying model inconsistency. Their findings indicate LLMs may be too inconsistent for deployment in high-stakes scenarios.

**Strengths:**

- Rigorous experimentation with BERTScore and good qualitative examples.

- Good comparison with synthetic perturbations of TruthQA.

**Weaknesses:**

- The point of the paper's main findings is unclear given the applications, and the authors' rebuttal further adds confusion by arguing non-determinism is favorable in wargames. What is being measured with BERTScore is not internal reliability. I would argue that the inconsistency being measured here isn't inherently bad. The authors really need a more nuanced metric capturing inconsistency in outcomes: in other words, they need to determine if multiple dissimilar decision paths lead to similarly or dissimilarly desirable outcomes.

- The writing of this paper is confusing. Despite the title, it seems like the authors focus primarily on justifying BERTScore as a consistency metric rather than wargame simulation, which becomes an afterthought. I think the paper would benefit from a broader focus, with more applications, comparison across different metrics of consistency, and deeper analysis of the implications.

- Lack of discussion around mitigation of undesirable inconsistency.

The core promise of measuring how LLM consistency impacts high stakes decision-making, particularly military decision-making, is a great idea. However, I believe the paper is far from being in a publishable state and is not appropriate for ICLR.

**Additional Comments On Reviewer Discussion:**

The reviewers are all leaning towards rejection. They did note the reproducibility of the work and focus on high-stakes real-world applications. However, there were concerns about lack of generalization to non-military scenarios (reviewers McSh, cN3C) and the sole focus on BERTScore without attempts to develop a metric specific to inconsistency measurement (reviewers McSh, xYfm). Reviewer cN3C also brought up a good point about observed inconsistency and the authors' choice of setting the temperature to 0 during generation, which led to a discussion reinforcing my take-away that inconsistency is ultimately not fine-grained enough and ill-defined in this work. What the authors really need to measure is the balance between unpredictability of decision-making and predicability (or desirableness) of outcomes.

---

### Decision · Program_Chairs · 2025-01-22

Reject